# Comparative Analysis of the Permutation and Multiscale Entropies for Quantification of the Brain Signal Variability in Naturalistic Scenarios

**DOI:** 10.3390/brainsci10080527

**Published:** 2020-08-06

**Authors:** Soheil Keshmiri

**Affiliations:** The Thomas N. Sato BioMEC-X Laboratories, Advanced Telecommunications Research Institute International (ATR), 2-2 Hikaridai Seika-cho, Kyoto 619-02, Japan; soheil@atr.jp

**Keywords:** differential entropy, multi-scale entropy, permutation entropy, whole-brain variability, brain information processing

## Abstract

As alternative entropy estimators, multiscale entropy (MSE) and permutation entropy (PE) are utilized for quantification of the brain function and its signal variability. In this context, their applications are primarily focused on two specific domains: (1) the effect of brain pathology on its function (2) the study of altered states of consciousness. As a result, there is a paucity of research on applicability of these measures in more naturalistic scenarios. In addition, the utility of these measures for quantification of the brain function and with respect to its signal entropy is not well studied. These shortcomings limit the interpretability of the measures when used for quantification of the brain signal entropy. The present study addresses these limitations by comparing MSE and PE with entropy of human subjects’ EEG recordings, who watched short movie clips with negative, neutral, and positive content. The contribution of the present study is threefold. First, it identifies a significant anti-correlation between MSE and entropy. In this regard, it also verifies that such an anti-correlation is stronger in the case of negative rather than positive or neutral affects. Second, it finds that MSE significantly differentiates between these three affective states. Third, it observes that the use of PE does not warrant such significant differences. These results highlight the level of association between brain’s entropy in response to affective stimuli on the one hand and its quantification in terms of MSE and PE on the other hand. This, in turn, allows for more informed conclusions on the utility of MSE and PE for the study and analysis of the brain signal variability in naturalistic scenarios.

## 1. Introduction

Recent neuroscientific findings present compelling empirical [1,2,3] and theoretical [4,5] evidence for the importance of the brain signal variability in its function. Such a variability stems from the interaction between neuronal circuits [6,7] across broad spatiotemporal scales [8,9]. This variability is further hypothesized to signify the cortical self-organized criticality [10,11,12,13] in which the brain capacity for information processing is maximized [14,15]. These observations help verify the potential role of entropy in identification of the brain signal variability [16,17,18,19,20,21] and the significance of such a variability in its function [22,23,24].

However, the entropy of biological signals can be drastically affected by such factors as long-range autocorrelation (due to reduction of entropy by correlation [25]), the signal length, and the presence of non-stationarity [26]. To mitigate these affects, a number of entropy estimators are introduced [27,28] among which multiscale entropy (MSE) [29] and permutation entropy (PE) [30] are widely used.

MSE utilizes the sample entropy [28] to quantify the signal variability at different time scales (through coarse graining process) [31,32]. It is successfully applied in the study of schizophrenia [33], depression [34], autism [35,36], and Alzheimer’s disease [37,38]. It also appears as a relibale tool for analysis of the effect of ageing on brain function [39,40] as well as the sleep cycle [41].

On the other hand, PE utilizes the reoccurrences of ordinal patterns in the signal to estimate its variability through the application of Shannon entropy [42]. PE is shown to be more robust to noise [30] and invariant under non-linear scale transformations [43]. Applications of PE include the difference in heart rate variability (HRV) between healthy individuals and the congestive heart failure (CHF) patients [44], detection of epileptic seizure [45], and the effect of anesthetic drugs on humans [46] and non-humans [47]. Zanin et al. present a review of PE applications.

### 1.1. The Purpose of Present Study

Within the domain of neuroscientific research, MSE and PE are primarily used in two specific scenarios: (1) the affect of brain pathology on its function [33,34,35,36,37,38,45] (2) the differential states of wakeful and unconscious brain [41,46,47,48]. This results in a lack of sufficient research on applicability of these measures in more naturalistic scenarios. Furthermore, the utility of these measures for quantification of the brain function and with respect to its entropy is not well studied. Such analyses are crucial for determining the level of correspondence between these measures on the one hand and the entropy of neurophysiological signals on the other hand. For instance, MSE utilizes several coarse-grained versions of original time series (referred to as scales) to refine the estimate of its variability [29]. In contrast, PE achieves this objective via the realization of the ordinal patterns in the time series [30]. However, the lack of comparative analysis between these measures with entropy limits their interpretability when used for quantification of the brain signal entropy.

The present study addresses these shortcomings through comparative analysis of MSE and PE for study of the human brain signal variability in response to naturalistic stimuli. To this end, it uses the Shanghai Jiao Tong University (SJTU) Emotion EEG Dataset (SEED) [49] that comprises human subjects’ sixty-two-channel EEG recordings. These EEG signals were acquired when the human subjects in the SEED study watched fifteen four-minute movie clips with positive, neutral, and negative contents.

### 1.2. Contributions

The contribution of the present study is threefold. First, it identifies a significant anti-correlation between MSE and entropy, whose affect is stronger in the case of negative than positive or neutral affect. Second, it verifies that MSE significantly differentiates between these affective states. In particular, it indicates that the negative affect is associated with lowest MSE, that the positive affect’s MSE is higher at the small-scale, and that the neutral affect is associated with higher MSE at the large-scale. These results highlight the level of association between the brain’s entropy in response to affective stimuli on the one hand and its quantification in terms of MSE and PE on the other hand.

These results help draw a level of association between the brain’s entropy in response to affective stimuli on the one hand and its quantification in terms of MSE and PE on the other hand. This, in turn, allows for more informed conclusions on the utility of these measures for study and analysis of the brain function.

## 2. Materials and Methods

### The Dataset

SEED [49] includes (Figure 1B) fifteen Chinese subjects’ sixty-two-channel EEG recordings (7 males and 8 females; Mean (M) = 23.27, Standard Deviation (SD) = 2.37). These recordings were acquired during the participants’ watching of fifteen four-minute Chinese movie clips with negative, neutral, and positive affective content (five clips per affect). The participants’ selection criteria were based on the Eysenck Personality Questionnaire (EPQ) [50] personality traits. This questionnaire analyzes the individuals’ personality along three independent temperament’s dimensions. They are neuroticism/stability, extraversion/introversion, and psychoticism/socialization.

The movie clips were selected based on three criteria: length (i.e., four-minute to prevent fatigue), content (easy to comprehend without requiring explicit explanations), and single-affect elicitation (e.g., only negative affect). To choose proper movie clips, the authors recruited twenty volunteers who were different from those who took part in SEED. These additional individuals assessed a pool of movie clips using the Likert five-point scale. The authors then used these assessments and selected 15 movie clips whose average scores were ≥3 and were also ranked among the top five in their respective affect category. The affectiveness of these selected movie clips were further verified in a follow-up study [52] that included nine participants (different from the twenty individuals who first rated them).

In each experiment, per participant, the movie clips were ordered to prevent two similar-affect clips (e.g., both having negative content) to follow one another. There was a five-second hint period before the start of each clip (i.e., similar to [52]). After watching each four-minute movie clip, the participants answered three questions that assessed their emotion-feeling states [51]. These questions included participants’ actually felt emotion while watching the clip, have they previously watched the full movie that the clip was taken from, and how well they comprehended the clip’s content. After answering these questions, the participants took a fifteen-second rest before the next movie clip started. Every participant took part in three experiments (one week apart). Every experiment used the same set of fifteen movie clips. All participants watched them in the same order of presentation.

SEED’s preprocessing steps were as follows. First, the EEG segments that corresponded to the duration of each movie clip were extracted (i.e., per EEG channel, per affect, per movie clip, and per participants). These segments were then down-sampled to 200 Hz sampling rate. Next, they were visually examined against the individuals’ accompanying electromyography (EMG) and electrooculography (EOG) recordings. Subsequently, the data that were contaminated by potential muscle- (i.e., EMG) or eye-movement (i.e., EOG) were manually removed. The EOG was further utilized to identify blink artifacts in these EEG recordings. To attenuate the noise (e.g., cardiac pulsations, respiration, etc.), the EEG signals were band-pass filtered between 0.3 to 50 Hz. Further details about SEED and its data preprocessing can be found in [49,52] and at http://bcmi.sjtu.edu.cn/~seed/seed.html.

## 3. Data Validation

The present study utilized the EEG recordings that pertained to the first participation of every individual and included all of their fifteen movie clips (i.e., five clips per affect) in their session (Figure 1A). For each individual, all sixty-two EEG channels were used. Prior to further analyses, the following EEG signals’ validation steps were applied.

To ensure that DE, MSE, and PE computations were not affected by the length of EEG data [32,53,54], all recordings were first confirmed to be sufficiently long (Mean (M) = 45,286.71, Standard Deviation (SD) = 2776.61, CI${}_{95\%}$ = [44,565.70 46,007.73], minimum = 37,000, maximum Length = 47,601) where CI${}_{95\%}$ stands for 95% confidence interval. These recordings were then trimmed to yield equal-length EEG time series (i.e., per channel, per affect, and per participant) as per the shortest available EEG recording (i.e., 37,000 data points).

Next, the participants’ EEG recordings were de-trended. This step was then followed by performing Augmented Dickey–Fuller (ADF) [55] and Kwiatkowski–Phillips–Schmidt–Shin (KPSS) [56] tests on them to ensure their covariance stationarity. These tests identified that one of the participant’s data did not satisfy covariance stationarity (i.e., trend-stationarity which implies mean and variance do not change over time). Consequently, this participant was excluded from further analyses. Additionally, the first session of another two participants’ EEG recordings did not also pass these tests. Therefore, their first sessions were replaced with their corresponding second and third sessions, respectively, that subsequently passed these tests.

## 4. Quantification of the EEG Channels’ Signal Variability Using DE, PE, and MSE

In the present study, DE, MSE, and PE were computed in terms of the brain variability across distributed regions. This approach was motivated by the findings that underlined the distributed nature of the brain function [57,58,59,60]. For instance, prior research indicated that most accurate neural signature of discrete emotions were associated with the brain’s anatomically distributed variational patterns [61]. It also identified that the use of joint activity from these regions more accurately discriminated between different emotions [62,63]. In this respect, Liu et al. [48] showed that MSE and FC were robust measures for mice spatiotemporal whole-brain variability across anesthetic and wakeful states. These results provided evidence for the involvement of large-scale cortical networks in the formation of high-level mental states and distinct emotions [64,65,66].

In line with these findings, Keshmiri et al. [67] also observed that DE of the brain’s distributed regions significantly distinguished between negative, neutral, and positive affect. Specifically, their results indicated that DE of the negative and positive affect were higher than that of neutral affect. This finding echoed the results that identified an increase in the brain’s information processing during the emotional than neutral contexts [68]. It was also in accord with the findings that indicated an increase in brain activity with attention [69,70]. Keshmiri et al. [67] also observed that DE was able to significantly differentiate between the negative, neutral, and positive affect. These results further supported the observations that considered distributed brain regions were involved in the formation of differential affect and emotions [66,71,72]. In light of these findings, the present study considered DE from [67] as a basis for evaluation of the utility of MSE and PE for quantification of the brain responses to negative, neutral, and positive affective stimuli.

### 4.1. The Procedure

The present study followed the same steps that were outlined in [67] to compute DE, PE, and MSE values for participants’ full-length EEG time series (i.e., 37,000 data points), per channel. For clarity, these steps are restated in this section.

For every trial, each participant’s data that pertained to each of its five movie clips (i.e., one EEG data file for each of the movie clip), per affect, were accessed, one-by-one, and their respective DE, MSE, and PE were computed. This step yielded five separate sets (i.e., one set for each of the movie clips, per affect), each of which containing DE, PE, and MSE values for each of sixty-two EEG channels, per participant. We then computed the average values of each of these entropic measures for each channel.

For example, in the case of PE computation for EEG channel FP1 (Figure 1B), there were five PEs (i.e., one PE for each of the movie clips of a given affect (e.g., negative affect): [PE${}_{movie\;clip1}^{FP1}$, PE${}_{movie\;clip2}^{FP1}$, PE${}_{movie\;clip3}^{FP1}$, PE${}_{movie\;clip4}^{FP1}$, PE${}_{movie\;clip5}^{FP1}$]. These five PEs were averaged (i.e., mean([PE${}_{movie\;clip1}^{FP1}$, PE${}_{movie\;clip2}^{FP1}$, PE${}_{movie\;clip3}^{FP1}$, PE${}_{movie\;clip4}^{FP1}$, PE${}_{movie\;clip5}^{FP1}$]). It is also apparent that these values’ order have no effect on their computed average PE for FP1 [73].

### 4.2. DE Computation

DE was computed using its non-parametric estimator [74] that is based on the nearest neighbour distance [75]. The present results utilized its implementation in [76].

In the case of DE, the computation step highlighted in Section 4 resulted in 1 × 62 vectors, per affect, per participant, where 62 refers to the number of EEG channels (Figure 1B).

### 4.3. PE Computation

PE computation was carried out using its original implementation by Bandt and Pompe [30]. In the case of PE, practical consideration for its use favours the choice of permutation order that is 3≤m!<<N [53,77,78] where m and N refer to permutation order and the length of the time series, respectively. Considering the length of EEG time series in the present study (i.e., 37,000 data points), the only suitable permutation orders were 3–7. Therefore, the main text reported PE results using order 3 and time lag 1 (i.e., its original implementation by Bandt and Pompe [30]). Appendices Appendix J, Appendix K and Appendix L provide the results pertinent to permutation orders 5, 6 and 7.

In the case of PE, computational steps in Section 4 resulted in 1 × 62 vectors (for all permutation orders 3, 5, 6, and 7), per affect, per participant, where 62 refers to the number of EEG channels (Figure 1B).

### 4.4. MSE Computation

The present study utilized the original MSE implementation by Goldberger et al. [32]. It also followed their guidelines on the choice of parameters, thereby using the scale factor 20 (MSE1 through MSE20 hereafter) and the similarity criterion 0.20 (i.e., a positive real value, typically within 10.0% to 20.0% of the time series standard deviation [32]).

MSE computation also followed the same computational steps in Section 4 (i.e., similar to DE and PE). Unlike DE and PE, MSE computation resulted in 20 × 62 matrices, per affect, per participant, where 62 refers to the number of EEG channels (Figure 1B) and 20 refers to MSE1–MSE20, per channel, per affect.

## 5. Analysis

For both PE and MSE, their Spearman correlations with DE, per EEG channel, per affect, were computed. This was followed by their bootstrap (10,000 simulation runs) correlations at 95.0% (i.e., *p* < 0.05) confidence interval. This test considered the null hypothesis

**Hypothesis** **1.**
*There was no correlation between participants’ PEs/MSEs and the corresponding DEs of their EEG channels.*


Hypothesis 1 was then tested against the alternative hypothesis

**Hypothesis** **2.**
*There was a significant correlation between participants’ PEs/MSEs and the corresponding DEs of their EEG channels.*


We further computed the p-value of these tests as a fraction of the distribution that was more extreme than the actually observed correlation values. This was carried out using a two-tailed test in which the correlation coefficients’ absolute values were used so that both, the positive and the negative correlations were accounted for.

Next, the significance of EEG channels’ PE and MSE for each of the negative, neutral, and positive affect were compared (i.e., with respect to DE) using Kruskal–Wallis test. This was then followed by posthoc Wilcoxon rank-sum tests between every pair of affect. These results were further verified through the application of the paired two-sample bootstrap test of significance (10,000 simulation runs) at 95.0% confidence interval on each pair of these affective states. These paired two-sample bootstrap tests considered the null hypothesis

**Hypothesis** **3.**
*The difference between PE/MSE values of participants’ EEG channels in two different mental states was non-significant.*


They tested Hypothesis 3 against the alternative hypothesis.

**Hypothesis** **4.**
*The PE/MSE values of participants’ EEG channels were significantly different in two different mental states.*


### Reported Results and Affect Sizes

In the case of two-tailed bootstrap tests, their mean, standard deviation, and 95.0% confidence interval are reported.

Results pertinent to MSE1 and MSE20, per participant, per affect, per EEG channel, are reported in the main text. Appendices Appendix G, Appendix H and Appendix I provide these results for MSE2–MSE19.

For PE, the main text includes its results based on permutation order 3. Subsequently, Appendices Appendix J, Appendix K and Appendix L present the results for permutation orders 5, 6, and 7.

Appendix M provide the Spearman Correlations Between MSE1 versus MSE2–MSE20. The rand p-values associated with these figures are shown in Table A17, Table A18 and Table A19.

The effect size for the Kruskal–Wallis test was computed using $r=\sqrt{\frac{\chi^{2}}{N}}$ [79] where *N* and $\chi^{2}$ are the sample size and the test-statistics. For Wilcoxon tests, the effect size $r=\frac{W}{\sqrt{N}}$ [80] was used. In this equation, *W* is the Wilcoxon statistics. All results were Bonferroni-corrected. All analyses were carried out in Matlab 2016a.

In regards to the adapted analyses by the present study, two points beg further clarification: the choice of non-parametric tests and the bootstrap test of significance. The choice of non-parametric tests were due to the fact that the computed DE, PE, and MSE, per affect, were not normally distributed (separately as well as combined, with respect to both: individuals and EEG channels). On the other hand, the use of bootstraps was motivated by the fact that the reported analyses were based on a small sample of fourteen individuals. As a result, it was necessary to ensure that the observed differences between DE, MSE, and PE were not due to a sub-sample of participants (i.e., skewed values). Applying bootstrap tests with large re-sampling successions (10,000 simulations) at 95.0% confidence interval (i.e., *p* < 0.05 significance level) allowed for further evaluation of the reported results.

## 6. Results

### 6.1. DE vs. PE

Figure 2 shows the correlations between participants’ DE and their corresponding PE values. PE and DE were significantly correlated in the case of negative affect (Figure 2A, r = 0.32, *p* < 0.001). However, such correlations were non-significant in the case of neutral (r = 0.05, *p* = 0.15) and positive (r = 0.05, *p* = 0.15) affect. These results were further supported by their corresponding bootstrap tests (10,000 simulation runs) at 95.0% confidence interval (Appendix A).

Figure 3A shows the grand averages of the spatial maps of participants’ PE in negative, neutral, and positive affect. Kruskal–Wallis test indicated a significant difference among participants’ positive, neutral, and negative affect (*p* < 0.001, H(2, 2603) = 20.79, r = 0.09). Posthoc Wilcoxon test (Figure 3B) indicated higher PE values in the case of positive than neutral affect (*p* < 0.001, W(1734) = 4.52, r = 0.11, M${}_{Positive}$ = 2.56, SD${}_{Positive}$= 0.04, M${}_{Neutral}$ = 2.54, SD${}_{Neutral}$= 0.05). Similarly, the negative affect was associated with higher PEs than the neutral affect (*p* < 0.01, W(1734) = 3.03, r = 0.07, M${}_{Negative}$ = 2.55, SD${}_{Negative}$ = 0.05). On the other hand, the difference between the PEs of the positive and negative affect were non-significant (*p* = 0.21, W(1734) = 1.26). However, these significant differences did not survive the two-sample bootstrap test of significance (10,000 simulation runs) at 95.0% confidence interval (Appendix B).

### 6.2. MSE1 vs. DE

Figure 4 shows the correlations between participants’ DEs and MSE1s. MSE1 and DE were significantly anti-correlated in the case of negative (Figure 4A, r = −0.50, *p* < 0.001), neutral (Figure 4B, r = −0.13, *p* < 0.001), and positive affect (Figure 4C, r = −0.13, *p* < 0.001). These results were further supported by their corresponding bootstrap tests (10,000 simulation runs) at 95.0% confidence interval (Appendix C).

Figure 5A shows the grand averages of the spatial maps of participants’ MSE1s in negative, neutral, and positive affect. These subplots identify incremental patterns of MSE1 from negative to positive affect. Kruskal–Wallis test on these MSE1 values identified a significant difference between these three affective states (*p* < 0.001, H(2, 2603) = 160.39, r = 0.25). Posthoc Wilcoxon test (Figure 5B) indicated that MSE1 in the case of positive affect was significantly higher than neutral (*p* < 0.001, W(1734) = 4.51, r = 0.11, M${}_{Positive}$ = 1.22, SD${}_{Positive}$ = 0.46, M${}_{Neutral}$ = 1.13, SD${}_{Neutral}$ = 0.38) and negative (*p* < 0.001, W(1734) = 11.84, r = 0.28, M${}_{Negative}$= 0.95, SD${}_{Negative}$= 0.45) affect. It also showed that neutral affect’s MSE1 was significantly higher than MSE1 associated with the negative affect (*p* < 0.001, W(1734) = 8.90, r = 0.22). These results were further supported by their corresponding two-sample bootstrap tests of significance (10,000 simulation runs) at 95.0% confidence interval (Appendix D).

### 6.3. MSE20 vs. DE

There were also significant anti-correlations between participants’ DEs and MSE20s (negative: Figure 6A, r = −0.56, *p* < 0.001, and neutral: Figure 6B, r = −0.41, *p* < 0.001, and positive: Figure 6C, r = −0.41, *p* < 0.001). These anti-correlations were stronger in the case of MSE20 than MSE1. They were further supported by their corresponding bootstrap tests (10,000 simulation runs) at the 95.0% confidence interval (Appendix E).

Figure 7A shows the grand averages of the spatial maps of participants’ MSE20s in negative, neutral, and positive affect. These subplots identify a pattern in which MSE20s associated with neutral affect is higher than negative and positive affect. Kruskal–Wallis on these MSE20s identified a significant difference between participants’ three affective states (*p* < 0.001, H(2, 2603) = 155.37, r = 0.24). However in this case and unlike MSE1, posthoc Wilcoxon test (Figure 7B) revealed that neutral affect’s MSE20 was significantly higher than positive ( *p* < 0.01, W(1734) = 2.84, r = 0.07, M${}_{Positive}$ = 1.26, SD${}_{Positive}$ = 0.33, M${}_{Neutral}$ = 1.31, SD${}_{Neutral}$ = 0.31) and negative (*p* < 0.001, W(1734) = 9.20, r = 0.22, M${}_{Negative}$= 1.09, SD${}_{Negative}$ = 0.38) affect. It also indicated that positive affect’s MSE20 was significantly higher (i.e., similar to MSE1) than the MSE20 of negative affect (*p* < 0.001, W(1734) = 11.85, r = 0.28). These results were further supported by their corresponding two-sample bootstrap tests of significance 10,000 simulation runs) at a 95.0% confidence interval (Appendix F).

## 7. Discussion

The present study sought to examine the utility of MSE and PE for study of the effect of naturalistic affective stimuli on brain signal variability. For this purpose, it utilized SEED [49]: a publicly available dataset of human subjects’ sixty-two-channel EEG recordings. These EEG signals were recorded while participants watched movie clips whose contents elicited negative, neutral, and positive affect. Our study was primarily motivated by two observations. First, previous research has mostly considered MSE and PE for investigation of either the effect of brain pathology on its function [33,34,35,36,37,38,45] or the altered state of consciousness [41,46,47,48]. This has left the utility of these measures for the study of brain variability in more naturalistic settings unanswered. Second, the lack of comparative analysis of MSE and PE with the brain signal entropy has further limited the scope of their interpretability when used for quantification of the brain signal entropy.

MSE appeared to complement the findings on the use of DE for quantification of the differential effect of negative, neutral, and positive affect on the brain activity. Specifically, Keshmiri et al. [67] identified that DE of the brain’s distributed regions significantly distinguished between negative, neutral, and positive affect. They also observed that DEs of negative and positive affect were higher than neutral’s DE. This finding echoed the brain’s higher information processing during the emotional, rather than neutral contexts [68]. In this respect, the present results found that positive and neutral affects were associated with higher MSE than the negative affect. They further indicated that the positive affect’s MSE was higher at the small-scale and the neutral affect was associated with higher MSE at the large-scale. These results were in accordance with the findings that maximum entropy does not necessarily promote higher signal variability [25,29]. They also extended the findings on the interplay between small and large scale MSEs [39,48,81,82] to the case of brain affective responses to naturalistic stimuli.

On the other hand, we found little evidence in support of the use of PE. For instance, the use of permutation order-3 resulted in a significant correlation between PE and DE in the case of negative affect. Although this correlation was in accord with the correspondence between PE and signal’s entropy [30,83,84], such a correlation was absent in the case of neutral and positive affect. Surprisingly, such a correlation was absent altogether in the case of higher-order permutations (Appendices Appendix J, Appendix K and Appendix L). Considering the present results, PE appeared to more readily associate with the brain variability that is characterized with distinct states (e.g., epileptic seizure [45], the affect of anesthetic drugs [46,47]) that presumably take place in a dichotomous fashion [66] than subtle changes in mental states during naturalistic settings (e.g., negative than neutral or positive affect).

A number of studies utilize PE to interpret the affect of such activities as meditation [85] and mediated social communication [86] on human subjects. These studies use PE to construe the observed changes in the brain activity in terms of relaxation and induced comfort in such settings. The present results challenged the plausibility of such interpretations. In particular, the observed insensitivity of PE to subtle changes in brain variability appeared to explain the discrepancy of the findings in the meditation literature. For instance, Vyšata et al. [85] reported a decrease in global permutation entropy during the mediation (interpreted as a sign for relaxation by the authors). On the contrary, Kakumanua et al. [87] observed an increase in PE that was only present in the case of experienced meditators. This was also the case in mediated social communication in which the lower [86] and higher [88,89] entropy was assumed to highlight such self-assessed feelings as relaxed mood, interest, and the feeling of human presence. Furthermore, the mediated social communication studies were based on a surprisingly limited number of brain sites (e.g., only two forehead sites in [86,88,89]). As a result, it was unclear whether the observed changes in entropy were due to the individuals’ interaction with the media or potentially a residue of overall brain signal variability [1,3] which may not necessarily pertain to the affect of stimuli. Most importantly, these results were inconsistent with the findings that underlined the substantial contrast between human–human and human–agent interactions (HHI and HAI, respectively) [90]. For example, Rauchbauer and colleagues [91] observed that neural markers of mentalizing [92,93] and social motivation [94] were only activated during HHI. These studies also suggested that although the brain activity within the person perception network (PPN) was not reduced during HAI (as compared to HHI), the activity within the theory-of-mind (TOM) network [95,96] was reduced [97,98]. These findings along with the insensitivity of PE to differential affect in the present study, appear to be at odds with the interpretation of decreased PE as a measure of relaxation/comfort in mediated communication [86].

Taken together, the present study implied that MSE and PE may yield different results in the case of naturalistic settings in comparison with such scenarios as the affect of brain pathology on its function [33,34,35,36,37,38,45] and altered states of consciousness [41,46,47,48]. These results help realize the level of association between entropy, on the one hand, and MSE and PE on the other hand, thereby allowing for more informed conclusions on their use for the quantification of the brain variability in response to naturalistic stimuli.

On the other hand, it is also crucial to emphasize that the present results suffer from two important limiting factors: small sample size and the lack of cultural diversity within this sample. In this regard, the accompanying bootstrap tests with large repeated re-sampling partially allowed for mitigating the potential impact that the small sample might have otherwise imposed on the analyses. However, such tests must not be interpreted as substitutes for larger sample size and therefore care must be taken to avoid improper generalization of the present findings. In the same vein, cultural diversity is the key to a thorough understanding of the full spectrum of the humans’ emotion and mental states. Although individuals’ ability to experience affect appears to be a unifying facet across cultures [99], this does not necessarily imply that all individuals experience and respond to affective stimuli in a similar way. Therefore, one must take caution to prevent unwarranted generalization of the present results across different cultures.

## 8. Limitations and Future Direction

An important characteristic of the brain function is the innate interactivity among its distributed regions [20]. This interplay among brain’s distributed regions implies the flow of information across its multiple areas whose information-content may concurrently contribute to a number of functional pathways [100]. The present study fell short in considering the potential effect of a such a flow of information on computed DE, PE, and MSE values. Therefore, future research to verify the possibility of such impacts [101,102] is necessary, thereby allowing for more informed conclusion on present results.

Our analyses identified significant anti-correlations between DE and MSE at different scales (i.e., MSE1–MSE20). They also indicated that such anti-correlations were stronger in the case of negative than positive and neutral affect. This differed from the results of the altered states of consciousness [48] in which positive/negative correlations were associated with the small/large scales. Although Liu et al. [48] also reported the weakening of correlations in the case of simulated brain variability, observed correlations in the present study were substantially weaker. These differences may suggest the differential behaviour of MSE in naturalistic settings compared to such scenarios as a study of the brain variability under anesthesia and wakefulness. On the other hand, the observed correlational differences might be due to the differing operational principle between DE that was used in the present study and the regional entropy (RE) [15,103] that Liu et al. [48] utilized. The comparative analysis of RE and DE is an interesting venue for future research to shed further light on their respective degree of correlation with MSE.

The Spearman correlation between DE and MSE1 appeared to suggest a degree of a nonlinear relationship between these two quantities. A potential reason behind the observed effect might be the shared information among different EEG channels with respect to their MSE1. The possibility of higher shared information between DE and MSE1 became more plausible, considering the increased anti-correlation between DE and MSE of higher scales (e.g., MSE20). This might also hint at the possibility of lower scale MSE (e.g., MSE1) to more closely reflect/capture the local information processing [39,81] at the sight of EEG channels. This, in turn, could imply that the changes in the lower scale MSE was more in line (i.e., less anti-correlated) with the signal entropy (i.e., its DE). It is apparent that these potential non-linear relations could not be captured by Spearman correlation (i.e., a linear measure). Therefore, it is crucial to further investigate the nature and potential causes of such variation in the observed anti-correlations through the use of measures that can quantify linear as well as nonlinear relations (e.g., mutual information (MI) [104]).

The small sample size is another limitation of the present work. This concerning issue [58,105,106,107] is best highlighted by Lindquist [66] that found 914 affect-related experimental contrasts included only 6827 participants (i.e., 6827/914 ≈ 7.47 individuals on average). This is further worsened by the fact that many of these studies comprise individuals from the same demographic background (however, see [108] for a small deviation). Although individuals’ ability to experience affect appears to be a unifying and common concept across cultures [109,110], the present results could be further enriched through the inclusion of larger human samples from different age groups and cultural backgrounds. This will, in turn, allow for a thorough comprehension and generalization of the utility of DE, MSE, and PE within the context of the brain variability in response to naturalistic stimuli.

Entropic measures have also been widely used as robust features for classification of the individuals’ mental [45,49]. Interestingly, Liu and colleagues [111] used a considerably large resting-state functional magnetic resonance imaging data from Human Connectome Project (998 individuals) to demonstrate that the information from cortical entropy profiles could effectively predict diverse facets of human subjects’ cognitive ability. Although the present results pointed at subtle differences between DE and MSE on the one hand, and PE on the other hand, they did not discard their potentials as robust features for quantification of the b374-382rain activity in such applications as brain–computer interface (BCI) [112] and neurofeedback brain training [113]. In this regard, an intriguing venue for future exploration is to consider the combination of these features to utilize their differential power for quantification of the brain variability in such applications.

The adapted non-parametric estimator of DE in the present study utilizes the nearest neighbour distance for its DE estimation [74], thereby achieving a more robust estimate of DE in comparison with histogram-based techniques [26]. It also bypasses the need for imposing an unwarranted distribution on data that is necessary when one opts for parametric (e.g., Gaussian distribution and its closed-form parametric DE computation) approaches [26]. From a computational perspective, the use of nearest neighbour distance by this approach highlights an interesting underlying similarity between DE on the one hand and PE and SE on the other hand. Specifically, DE and PE computations share a basic principle in that they both estimate the entropy of a continuous time-series through its discretization: i.e., neighbourhood formation in the case of DE estimator and the signal’s symbolic representation in the case of PE. In the same vein (although in a much-restricted sense), DE and PE both relate to SE computation through the realization that the latter is based on a discretized (a binary discretization in this case) summary statistics of a given continuous time series. In this respect, it would be interesting for future research to further examine the effect of such discretization strategies on the level of correspondence between the estimated entropy by these algorithms. Another possibility for future research that is worth investigating would be to reevaluate the use of PE for estimation of the brain variability in response to naturalistic stimuli based on its more recent variant i.e., multiscale permutation entropy (MPE) [114].

### Ethics Statement

This study was carried out in accordance with the recommendations of the ethical committee of the Advanced Telecommunications Research Institute International (ATR). The protocol was approved by the ATR ethical committee (approval code: 0221200707001).

## Figures and Tables

**Figure 1 brainsci-10-00527-f001:**
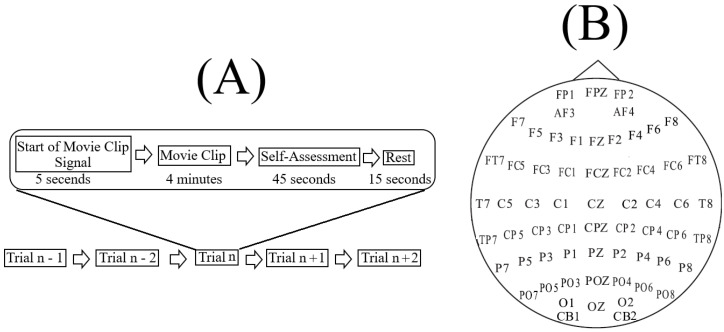
(**A**) Shanghai Jiao Tong University (SJTU) Emotion EEG Dataset (SEED) [49] paradigm. Each setting, per participant, comprised fifteen four-minute movie clips (i.e., n = 15). There was a five-second hint before the start of each clip. After each clip, participants answered (Likert 5-point scale) three questions that assessed their emotion-feeling states [51]. These questions included participants’ actually felt emotion while watching the clip, have they previously watched the fully movie that the clip corresponded to, and how well they comprehended the clip’s content. (**B**) Sixty-two EEG channels’ arrangement.

**Figure 2 brainsci-10-00527-f002:**
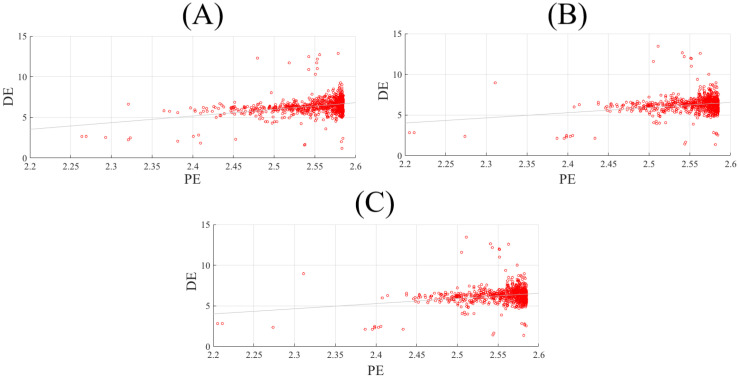
Spearman correlation between participants’ DEs (i.e., 62 DEs associated with 62 EEG channels of a participant, per affect) and their corresponding PE values (i.e., 62 permutation entropies (PE) associated with 62 EEG channels of a participant, per affect) in (**A**) negative, (**B**) neutral, and (**C**) positive.

**Figure 3 brainsci-10-00527-f003:**
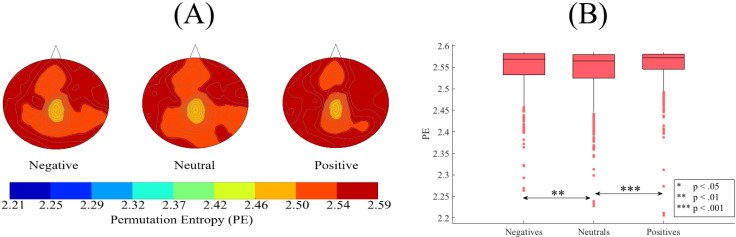
(**A**) Grand averages of the spatial maps of participants’ PEs in negative, neutral, and positive affect states. These subplots identify that PEs associated with these mental states exhibited highly resembling patterns that were approximately uniformly distributed over all brain regions. (**B**) Descriptive statistics of the posthoc Wilcoxon rank-sum tests applied on participants’ whole-brain PEs (i.e., 62 PEs associated with 62 EEG channels of a participant, per affect) in negative, neutral, and positive affect states. The asterisks mark the significant differences in these subplots.

**Figure 4 brainsci-10-00527-f004:**
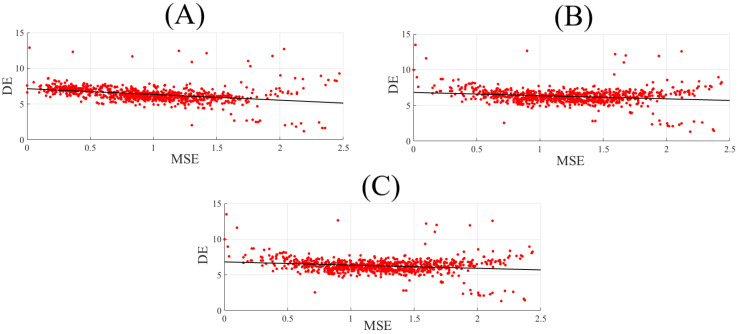
Spearman correlation between participants’ DEs (i.e., 62 DEs associated with 62 EEG channels of a participant, per affect) and their corresponding multiscale entropy (MSE)1 values (i.e., 62 MSE1s associated with 62 EEG channels of a participant, per affect) in (**A**) Negative (**B**) Neutral (**C**) Positive.

**Figure 5 brainsci-10-00527-f005:**
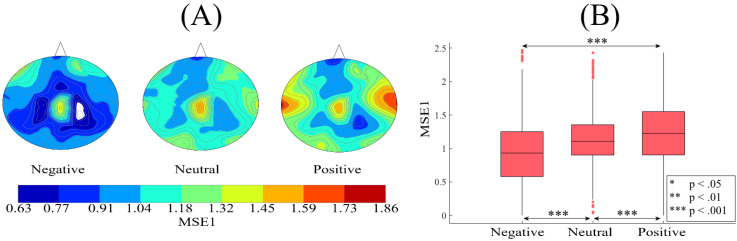
(**A**) Grand averages of the spatial maps of participants’ MSE1s (i.e., 62 MSE1s associated with 62 EEG channels of a participant, per affect) in negative, neutral, and positive mental states. Distribution of MSE1s in these spatial maps corresponds to the EEG channels’ arrangement in Figure 1B (i.e., one MSE1 value, per EEG channel) Differential patterns of the participants’ MSE1s that show an increase from negative to positive states is evident in these subplots. (**B**) Descriptive statistics of participants’ MSE1 in negative, neutral, and positive mental states. The asterisks mark the significant differences in these subplots.

**Figure 6 brainsci-10-00527-f006:**
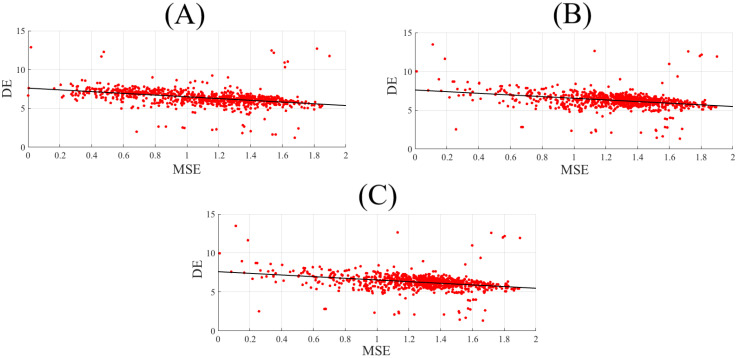
Spearman correlation between participants’ DEs (i.e., 62 DEs associated with 62 EEG channels of a participant, per affect) and their corresponding MSE20 values (i.e., 62 MSE20s associated with 62 EEG channels of a participant, per affect) in (**A**) negative, (**B**) neutral, and (**C**) positive.

**Figure 7 brainsci-10-00527-f007:**
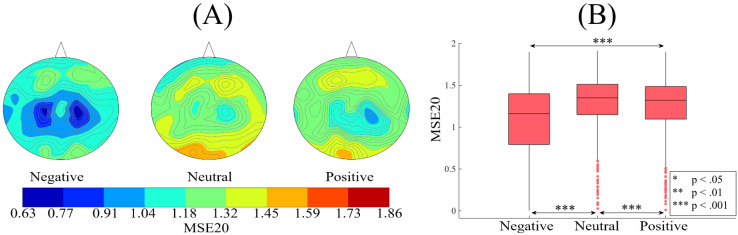
(**A**) Grand averages of the spatial maps of participants’ MSE20s (i.e., 62 MSE20s associated with 62 EEG channels of a participant, per affect) in negative, neutral, and positive mental states. Distribution of MSE20s in these spatial maps corresponds to the EEG channels’ arrangement in Figure 1B (i.e., one MSE20 value, per EEG channel). These subplots identify a pattern in which MSE20s associated with the neutral state is higher than the negative as well as the positive state. Similar to participants’ MSE1s, the negative state’s MSE20s are below those of the neutral and the positive states. (**B**) Descriptive statistics of the posthoc Wilcoxon rank-sum tests applied on participants’ whole-brain MSE20s (i.e., 62 MSE20s associated with 62 EEG channels of a participant, per affect).

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
