# Peer review of "Comparative Analysis of the Permutation and Multiscale Entropies for Quantification of the Brain Signal Variability in Naturalistic Scenarios"

_brainsci, 2020, doi:10.3390/brainsci10080527_

Round 1

Reviewer 1 Report

The muestral space has little variability with respect to age and the cultural level is not identified. This is important for analyzing the brain's responses with age deterioration.

Due to the dispersion of the data obtained with DE, MSE and PE, it seems that a multimodal study that would combine the responses of all of them using, for example, neural networks or fu would have been included.

In any case, the final study with 14 subjects is limited. It is recommended to increase the tests with more subjects, expanding the age range, explaining cultural origin.
However, the article makes a very interesting statistical study and it is recommended to empirically base the discussion.
Its publication would be interesting taking into account these recommendations.

Author Response

First and foremost, the author would like to take this opportunity to express his gratitude for the reviewer’s time and kind consideration to review the present manuscript. The reviewer’s comments substantially improved the quality of the present study and its presentation.

In what follows, point-by-point responses to the reviewer’s comments and concerns are provided.

Sincerely,

Reviewer 1

Reviewer’s Comment: The muestral space has little variability with respect to age and the cultural level is not identified. This is important for analyzing the brain's responses with age deterioration.

Author’s Response: The cultural background and the age of participants are provided in Section 2.1. The Dataset, lines 74-75, in the current version of manuscript. This information reads as follows.

“SEED [49] includes (Figure 1 (B)) fifteen Chinese subjects’ sixty-two-channel EEG recordings (7 males and 8 females; Mean (M) = 23.27, Standard Deviation (SD) = 2.37).

To further highlight the limitations imposed by small sample size, lack of cultural diversity, as well as the age of the participants, the following is discussed in Section 8. Limitations and Future Direction, lines 374-382, in the current version of manuscript.

“The small sample size is another limitation of the present work. This concerning issue [107-110] is best highlighted by Lindquist [66] that found 914 affect-related experimental contrasts included only 6827 participants (i.e 6827/914  7.47} individuals on average). This is further worsened by the fact that many of these studies comprise individuals from the same demographic background (however, see [111] for a small deviation). Although individuals' ability to experience affect appears to be a unifying and common concept across cultures [112,113], the present results could be further enriched through inclusion of larger human samples from different age groups and cultural backgrounds. This will, in turn, allow for a thorough comprehension and generalization of the utility of DE, MSE, and PE within the context of the brain variability in response to naturalistic stimuli.

Reviewer’s Comment: Due to the dispersion of the data obtained with DE, MSE and PE, it seems that a multimodal study that would combine the responses of all of them using, for example, neural networks or fu would have been included.

Author’s Response: This is indeed an interesting observation by the reviewer. To highlight this matter, the following discussion is added to Section 8. Limitations and Future Direction, lines 383-392, in the current version of the manuscript.

“Entropic measures have also been widely used as robust features for classification of the

individuals’ mental [45,116]. Interestingly, Liu and colleagues [117] used a considerably large

resting-state functional magnetic resonance imaging data from Human Connectome Project (998

individuals) to demonstrate that the information from cortical entropy profiles could effectively

predict diverse facets of human subjects’ cognitive ability. Although the present results pointed at

subtle differences between DE and MSE on the one hand and PE on the other hand, they did not

discard their potentials as robust features for quantification of the brain activity in such applications as brain-computer interface (BCI) [118] and neurofeedback brain training [119]. In this regard, an intriguing venue for future exploration is to consider the combination of these features to utilize their differential powers for quantification of the brain variability in such applications.”

Reviewer’s Comment: In any case, the final study with 14 subjects is limited. It is recommended to increase the tests with more subjects, expanding the age range, explaining cultural origin.

Author’s Response: The present study used an open-access database of human subjects EEG recordings (reference [49] in the current version of the manuscript). This choice was made to increase the possibility of reproducing the results. Unfortunately, the author is unable to increase the number of participants at present. However, the author certainly appreciates the importance of the issue raised by the reviewer. This issue is highlighted in Section 8. Limitations and Future Direction, lines 374-382, in the current version of manuscript and also in the author’s response to the reviewer’s comment The muestral space has little variability with respect…”

Reviewer 2 Report

The paper is good in quality, and it has a capacity of attracting readers. The following suggestions may be considered to enhance the quality and clarity of the manuscript.
1. Manuscript needs revision both at the level of concepts and write up.
2. The abstract of the paper needs to be rewritten and the major contribution of the paper need to be highlighted. The authors have not mentioned the problem statement, to carry out this research.

3. The introduction of the paper contains methodology steps that need to be moved to the proposed methodology portion instead of the introduction. The introduction needs rearrangement of ideas to improve the readability and purpose of this paper.

4. On page 8, section 5. this sentence has been repeated twice. "We reported the mean, standard deviation, and 95.0% confidence interval for these tests". It should be corrected.

5. The paper also has grammatical issues that need to be addressed.  There are too many grammatical mistakes in the manuscript. It should be corrected for the reader’s better understanding. Like, the word 'we' has been repeated again and again, which should be removed. Also, on page 12, section 7, this sentence should be corrected "Furthermore, they indicated that whereas the positive affect yielded the highest small-scale MSE, the neutral affect’s MSE was highest in the large-scale". The word 'whereas' should be removed for readers better understandability. unwantedly, this word is also repeated several times. It should be corrected.

7. A strong discussion on the results is required.

8. There are several mistakes in sentence formation as well as grammar.

9. Elaborate all tables briefly. 

10. The submission was not written well in many aspects including format and wording.

Author Response

First and foremost, the author would like to take this opportunity to express his gratitude for the reviewer’s time and kind consideration to review the present manuscript. The reviewer’s comments substantially improved the quality of the present study and its presentation.

In what follows, point-by-point responses to the reviewer’s comments and concerns are provided.

Sincerely,

Reviewer 2

Reviewer’s Comment: 1. Manuscript needs revision both at the level of concepts and write up.

Author’s Response: The current version of the manuscript has been thorough audited and its content has been modified accordingly. Please see the responses to the reviewer’s comments below.

Reviewer’s Comment: 2. The abstract of the paper needs to be rewritten and the major contribution of the paper need to be highlighted. The authors have not mentioned the problem statement, to carry out this research.

Author’s Response: The abstract is rewritten to reflect the rationale behind the present study and its contributions. The current version of the Abstract reads as follows.

“As alternative entropy estimators, multiscale entropy (MSE) and permutation entropy (PE) are utilized for quantification of the brain function and its signal variability. In this context, their applications are primarily focused on two specific domains: 1) the effect of brain pathology on its function 2) the study of altered states of consciousness. As a result, there is a paucity of research on applicability of these measures in more naturalistic scenarios. In addition, the utility of these measures for quantification of the brain function and with respect to its signal entropy is not well studied. These shortcomings limit the interpretability of these measures when used for quantification of the brain signal entropy. The present study addresses these limitations by comparing MSE and PE with entropy of human subjects' EEG recordings who watched short movie clips with negative, neutral, and positive contents. The contribution of the present study is threefold. First, it identifies a significant anti-correlation between MSE and entropy. In this regards, it also verifies that such an anti-correlation is stronger in the case of negative than positive or neutral affect. Second, it finds that MSE significantly differentiates between these three affective states. Third, it observes that the use of PE does not warrant such significant differences. These results highlight the level of association between brain's entropy in response to affective stimuli on the one hand and its quantification in terms of MSE and PE on the other hand. This, in turn, allows for more informed conclusions on the utility of MSE and PE for the study and analysis of the brain signal variability in naturalistic scenarios.

Reviewer’s Comment: 3. The introduction of the paper contains methodology steps that need to be moved to the proposed methodology portion instead of the introduction. The introduction needs rearrangement of ideas to improve the readability and purpose of this paper.

Author’s Response: The Introduction is audited and its content as well as its layout is modified. First, its writing is modified to increase its readability. Second, two new subsections are added. They are as follow.

  • 1. The Purpose of Present Study (lines 43-59, in the current version of the manuscript). This subsection highlights the shortcomings of the previous research with regards to the use of entropic measures in the study and analysis of the brain activity. It then summarizes how the present study attempts to address these shortcomings. The content of this subsection reads as follows.

“Within the domain of neuroscientific research, MSE and PE are primarily used in two specific scenarios: (1) the effect of brain pathology on its function [33-38,45] (2) the differential states of wakeful and unconscious brain[41,46-48]. This results in lack of sufficient research on applicability of these measures in more naturalistic scenarios. Furthermore, the utility of these measures for quantification of the brain function and with respect to its entropy is not well studied. Such analyses are crucial for determining the level of correspondence between these measures on the one hand and the entropy of neurophysiological signals on the other hand. For instance, MSE utilizes several coarse-grained versions of original time series (referred to as scales) to refine the estimate of its variability [29]. In contrast, PE achieves this objective via realization of the ordinal patterns in the time series[30]. However, the lack of comparative analysis between these measures with entropy limits their interpretability when used for quantification of the brain signal entropy.

The present study addresses these shortcomings through comparative analysis of MSE and PE for study of the human brain signal variability in response to naturalistic stimuli. To this end, it uses the Shanghai Jiao Tong University (SJTU) Emotion EEG Dataset (SEED) [49] that comprises human subjects' sixty-two-channel EEG recordings. These EEG signals were acquired when the human subjects in SEED study watched fifteen four-minute movie clips with positive, neutral, and negative contents.

  • 2. Contributions (lines 60-71, in the current version of the manuscript). In this subsection, the contributions of the present study are highlighted as follow.

“The contribution of the present study is threefold. First, it identifies a significant anti-correlation between MSE and entropy whose effect is stronger in the case of negative than positive or neutral affect. Second, it verifies that MSE significantly differentiates between these affective states. In particular, it indicates that the negative affect associates with lowest MSE, that the positive affect's MSE is higher at small-scale, and that the neutral affect is associated with higher MSE at large-scale. These results highlight the level of association between brain's entropy in response to affective stimuli on the one hand and its quantification in terms of MSE and PE on the other hand.

In addition, the part that did not fit the Introduction is moved to Section 4. Quantification of the EEG Channels’ Signal Variability Using DE, PE, and MSE, lines 128-147, in the current version of the manuscript. Its content reads as follow.

“In the present study, DE, MSE, and PE were computed in terms of the brain variability across distributed regions. This approach was motivated by the findings that underlined the distributed nature of the brain function [57-60]. For instance, prior research indicated that most accurate neural signature of discrete emotions were associated with the brain's anatomically distributed variational patterns [61]. It also identified that the use of joint activity from these regions more accurately discriminated between different emotions [62,63]. In this respect, Liu et al. [48] showed that MSE and FC were robust measures for mice spatiotemporal whole-brain variability across anesthetic and wakeful states. These results provided evidence for the involvement of large-scale cortical networks in formation of high-level mental states and distinct emotions [64-66].

In line with these findings, Keshmiri et al. [67] also observed that DE of the brain's distributed regions significantly distinguished between negative, neutral, and positive affect. Specifically, their results indicated that DE of the negative and positive affect were higher than that of neutral affect. This finding echoed the results that identified an increase in brain's information processing during the emotional than neutral contexts [73]. It was also in accord with the findings that indicated an increase in brain activity with attention [69,70]. Keshmiri et al. [67] also observed that DE was able to significantly differentiate between the negative, neutral, and positive affect. These results further supported the observations that considered distributed brain regions were involved in formation of differential affect and emotions[66,71,72]. In light of these findings, the present study considered DE from67 as a basis for evaluation of the utility of MSE and PE for quantification of the brain responses to negative, neutral, and positive affective stimuli.

Subsequently, a new subsection (Section 4.1. Procedure, lines 148-162, in the current version of manuscript) is added in which the overall DE, MSE, and PE computation is highlighted. This subsection also includes an example to illustrate DE, MSE, and PE computation (using PE as an example). The content of this subsection is the modified version of its content in the first draft of the manuscript.

Reviewer’s Comment: 4. On page 8, section 5. this sentence has been repeated twice. "We reported the mean, standard deviation, and 95.0% confidence interval for these tests". It should be corrected.

Author’s Response: a new subsection is added to Section 5. Analysis (Section 5.1. Reported Results and Effect Sizes, lines 212-231, in the current version of the manuscript). The concern sentence is then moved to this section (the first sentence). Furthermore, Section 5. Analysis is thoroughly checked to ensure that this sentence is repeated in its content.

Reviewer’s Comment: 5. The paper also has grammatical issues that need to be addressed.  There are too many grammatical mistakes in the manuscript. It should be corrected for the reader’s better understanding. Like, the word 'we' has been repeated again and again, which should be removed. Also, on page 12, section 7, this sentence should be corrected "Furthermore, they indicated that whereas the positive affect yielded the highest small-scale MSE, the neutral affect’s MSE was highest in the large-scale". The word 'whereas' should be removed for readers better understandability. unwantedly, this word is also repeated several times. It should be corrected.

Author’s Response: In the current version of manuscript, “we” is replaced with such phrases as “The current study,” “the present study.” The exceptions include:

  • Line 156-157, in the current version of the manuscript. “We then computed the average values of each of these entropic measures for each channel.
  • Line 197-198, in the current version of the manuscript. “We further computed the p-value of these tests as a fraction of the distribution that was more extreme than the actually observed correlation values.
  • Line 304-305, in the current version of the manuscript. “On the other hand, we found little evidence in support of the use of PE. For instance, the use of permutation order 3 resulted in a significant correlation between PE and DE in the case of negative affect.
  • Line 482484-305, in the current version of the manuscript. “We observed that these correlations were non-significant in all three mental states: negative (r = -.012, p = .71), neutral (r = -.053, p = .11), positive (r = -.048, p = .16).

The sentence "Furthermore, they indicated that whereas the positive affect yielded the highest small-scale MSE, the neutral affect’s MSE was highest in the large-scale" is replaced with the following (Section 7. Discussion, lines 298-300, in the current version of the manuscript).

“In this respect, the present results found that positive and neutral affect were associated with higher MSE than the negative affect. They further indicated that the positive affect's MSE was higher at small-scale and the neutral affect was associated with higher MSE at large-scale.

In addition, the manuscript has been checked for occurrences of “whereas” and its improper uses has been modified/corrected.

Reviewer’s Comment: 7. A strong discussion on the results is required.

Author’s Response: Sections 7. Discussion and 8. Limitations and Future Direction are further modified to more clearly reflect the results reported in the manuscript. In addition, a new content is added to Sections 7. Discussion that further elaborates on the caveats of the results reported by mediated communication based on entropic measures. It reads as follows (lines 321-335, in the current version of the manuscript).

“This was also the case in mediated social communication in which the lower [87] and higher [90,91] entropy wasassumed to highlight such self-assessed feelings as relaxed mood, interest, and the feeling of human presence. Furthermore, the mediated social communication studies were based on a surprisingly limited number of brain sites (e.g., only two forehead sites in [87,90,91] As a result, it was unclear whether the observed changes in entropy were due to the individuals' interaction with the media or potentially a residue of overall brain signal variability [1,3] which may not necessarily pertain to the effect of stimuli. Most importantly, these results were inconsistent with the findings that pinpointed the substantial contrast between human-human and human-agent interactions (HHI and HAI, respectively) [92]. For example, Rauchbauer and colleagues [93] observed that neural markers of mentalizing [94,95] and social motivation [96] were only activated during HHI. These studies also suggested that although the brain activity within the person perception network (PPN) was not reduced during HAI (as compared to HHI), the activity within the theory-of-mind (TOM) network [97,98] was reduced [99,100]. These findings along with the insensitivity of PE to differential affect in the present study appear to be at odds with the interpretation of decreased PE as a measure of relaxation/comfort in mediated communication [87].

Moreover, the following discussion is added to Section 8. Limitations and Future Direction, lines 383-392, in the current version of the manuscript.

“Entropic measures have also been widely used as robust features for classification of the

individuals’ mental [45,116]. Interestingly, Liu and colleagues [117] used a considerably large

resting-state functional magnetic resonance imaging data from Human Connectome Project (998

individuals) to demonstrate that the information from cortical entropy profiles could effectively

predict diverse facets of human subjects’ cognitive ability. Although the present results pointed at subtle differences between DE and MSE on the one hand and PE on the other hand, they did not discard their potentials as robust features for quantification of the brain activity in such applications as brain-computer interface (BCI) [118] and neurofeedback brain training [119]. In this regard, an intriguing venue for future exploration is to consider the combination of these features to utilize their differential powers for quantification of the brain variability in such applications.”

Reviewer’s Comment: 8. There are several mistakes in sentence formation as well as grammar.

Author’s Response: The entire manuscript is thoroughly audited and all mistakes are corrected.

Reviewer’s Comment: 9. Elaborate all tables briefly. 

Author’s Response: The summary for all tables that are reported in the Appendices are included within the content of each Appendix. These contents draw connections between each table and its respective figure that illustrates the result of corresponding bootstrap test. They can be found (in the current version of the manuscript) at lines

  • 422-425
  • 431-435
  • 438-441
  • 446-449
  • 456-459
  • 464-467
  • 475-479
  • 483-486
  • 495-498
  • 507-511
  • 517-520
  • 529-533
  • 539-542
  • 551-555

Reviewer’s Comment: 10. The submission was not written well in many aspects including format and wording.

Author’s Response: Please kindly refer to the responses to the reviewer’s comments above. The author would be glad to apply any further modification that the reviewer may consider necessary. Thank you.

Round 2

Reviewer 1 Report

I suggest trying to increase the case studies to validate the statistical study. In any case, if this is not possible, as is indicated by the authors, the discussion and conclusions should emphasize a heuristic explanation of the data.

Author Response

Reviewer’s Comment: I suggest trying to increase the case studies to validate the statistical study. In any case, if this is not possible, as is indicated by the authors, the discussion and conclusions should emphasize a heuristic explanation of the data.

Author’s Response: The following paragraph is added to Section 7. Discussion, lines 342-352, in the current version of manuscript.

“On the other hand, it is also crucial to emphasize that the present results suffer from two important limiting factors: small sample size and the lack of cultural diversity within this sample. In this regard, the accompanying bootstrap tests with large repeated re-sampling partially allowed for mitigating the potential impact that the small sample might have otherwise imposed on the analyses. However, such tests must not be interpreted as substitutes for larger sample size and therefore care must be taken to avoid improper generalization of the present findings. In the same vein, cultural diversity is the key to thorough understanding of the full spectrum of the humans’ emotion and mental states. Although individuals’ ability to experience affect appears to be a unifying facet across cultures [101], this does not necessarily imply that all individuals experience and respond to affective stimuli in a similar way. Therefore, one must take caution to prevent unwarranted generalization of the present results across different cultures.